# Learning to predict visual brain activity by predicting future sensory states

**Marcio Fonseca**
Directorate of Technology and Innovation
Chamber of Deputies
Brasília, BR 70160-900
`marcio.fonseca@camara.leg.br`

## Abstract

Deep predictive coding networks are neuroscience-inspired unsupervised learning models that learn to predict future sensory states. We build upon the *PredNet* implementation by Lotter, Kreiman, and Cox (2016) to investigate if predictive coding representations are useful to predict brain activity in the visual cortex. We use representational similarity analysis (RSA) to compare PredNet representations to functional magnetic resonance imaging (fMRI) and magnetoencephalography (MEG) data from the Algonauts Project (Cichy et al., 2019). In contrast to previous findings in the literature (Khaligh-Razavi & Kriegeskorte, 2014), we report empirical data suggesting that unsupervised models trained to predict frames of videos without further fine-tuning may outperform supervised image classification baselines in terms of correlation to spatial (fMRI) and temporal (MEG) data.

## 1   Introduction

Currently, convolutional neural networks trained on image recognition tasks are the best performing models to account for brain activity during visual object recognition (Schrimpf et al., 2018). The performance of such supervised models on recent benchmarks led to the idea that supervised learning may be a requirement to explain visual cortex activity, especially in higher cortical areas (Khaligh-Razavi & Kriegeskorte, 2014). In this work, we report experimental results suggesting that predictive coding models trained on unlabeled video data may outperform supervised baselines, yielding internal representations with higher correlation to representation dissimilarity matrices (RDM) (Kriegeskorte, Mur, & Bandettini, 2008) obtained from human fMRI and MEG data.

This paper summarizes our two main contributions. First, we find that a small predictive coding model trained on 4 hours of unlabeled video outperforms supervised baselines (He, Zhang, Ren, & Sun, 2016; Krizhevsky, Sutskever, & Hinton, 2012; Kubilius et al., 2018) in terms of correlation to RDMs across two fMRI and MEG datasets (Cichy et al., 2019). Moreover, we show that as we train the model on additional videos from the Moments in Time dataset (Monfort et al., 2018), the model internal representations become more similar to brain activity. Thus, our results suggest a promising avenue to build better models of human visual processing by scaling the training to bigger models and larger amounts of unlabeled video data.

Our work share similar methods with the *human-model similarity (HMS)* proposed by Blanchard, Kinnison, RichardWebster, Bashivan, and Scheirer (2019) as they use the same PredNet implementation and representational similarity analysis to measure the correlation between predictive coding representations and human fMRI. However, their work addresses the use of RDMs to inform hyperparameter search and model generalization. We focus on the comparison of the unsupervised

---

Code is available at `https://github.com/thefonseca/algonauts`

predictive coding model versus supervised baselines and the effect of scaling unsupervised training on the correlation scores.

## 2 Methods

**Predictive coding networks**   We build on the *PredNet* implementation by Lotter et al. (2016), which was shown to perform well on unsupervised learning tasks using video data. Inspired by the predictive coding theory (Friston & Kiebel, 2009), their model relies on the idea that to predict the next video frame, a model needs to capture latent structure that explains the image sequences. The PredNet architecture consists of recurrent convolutional layers (Xingjian et al., 2015) that propagate bottom-up prediction errors which are used by the upper-level layers to generate new predictions. For implementation details, please refer to the PredNet architecture description by Lotter et al. (2016).

**Unsupervised training**   We evaluate predictive coding models trained on different quantities of unlabeled videos (Table 1). The main idea is that the more data we use to train the model, the more "common sense" it should get about how events unfold in the world and, as a consequence, it should be better at disentangling latent explanatory factors. Using as starting point a PredNet pre-trained on the KITTI dataset (Geiger, Lenz, Stiller, & Urtasun, 2013), we further train the model with unlabeled videos from the Moments in Time dataset (Monfort et al., 2018), a large-scale activity recognition dataset. Additionally, we report correlation scores for a PredNet with random weights and a version trained from scratch using just videos from the Moments in Time dataset. The predictive coding

Table 1: Pre-training datasets used across the experiments. Action categories from the Moments in Time dataset are reported for reference and are not used during PredNet training.

| Pre-training dataset | Hours of video | Frames/Images | Categories |
|---|---|---|---|
| ImageNet | - | $\approx 1.2M$ | 1000 |
| KITTI | $\approx 1h$ | $\approx 41K$ | - |
| KITTI + Moments in Time | $\approx 4h$ | $\approx 160K$ | 10 |
| Moments in Time | $\approx 6h$ | $\approx 240K$ | 20 |

model is trained in an unsupervised way to predict the next frames using a top-down generative model. The errors between predictions and the actual frames are propagated bottom-up to update the prior for new predictions. In terms of architecture, we follow the same hyperparameter settings used in the original PredNet implementation proposed by Lotter et al. (2016), with four modules (PredNet-4) consisting of $3 \times 3$ convolutional layers with 3, 48, 96, and 192 filters and input frames with dimensions $128 \times 160$. We also train a larger 5-layer model (PredNet-5) with 3, 48, 96, 192, and 192 filters and input frames with a higher resolution of $256 \times 256$ pixels. The videos are subsampled at ten frames per second, and the network input is a sequence of ten frames for which the model generates ten frame predictions.

**Brain data**   Feature extraction and evaluation are performed on two datasets consisting of 92 silhouette object images Cichy, Pantazis, and Oliva (2014) and 118 object images on natural background (Cichy, Khosla, Pantazis, Torralba, & Oliva, 2016) as well as brain data recorded using fMRI (EVC and IT) and MEG (early and late in time) in response to visual stimuli. These datasets are curated to include representative visual categories such as animate/inanimate, human/non-human, and face/non-face, which are categorical divisions observed in high-level cortical areas in primates (Kriegeskorte, Mur, Ruff, et al., 2008).

**Feature extraction**   Each layer $l$ and timestep $t$ of the PredNet model has representation units $R_l^t$, which are extracted as features to be compared to brain data. No further preprocessing or dimensionality reduction is performed. Since the Algonauts datasets consists of images and not videos, we repeat each image ten times to make the input compatible with the PredNet architecture.

Extracted features are then transformed to representational dissimilarity matrices (RDM) as described by Kriegeskorte, Mur, and Bandettini (2008). RDM serves as a common space to compare representations of different models and capture the dissimilarity (1 minus Pearson correlation) of internal

representations generated for each pair of images in the dataset. To create RDMs from features, we use the Python implementation from the development kit provided by Cichy et al. (2019).

**Evaluation** The resulting RDMs for all model variants are compared to the RDMs from different brain regions, namely fMRI data from the early visual cortex (EVC) and the inferior temporal cortex (IT), and also MEG data for early and late stages of visual processing. The similarity of RDMs is computed in terms of Spearman's correlation, as defined by Kriegeskorte, Mur, and Bandettini (2008) and implemented in the Algonauts Python development kit. The reported correlation scores are normalized by the *noise ceiling*, which is the average of the correlations of each subject's RDMs to the average RDM across 15 subjects. Thus, the subject-averaged RDM is assumed to be the best estimate of an RDM generated by an ideal model (Cichy et al., 2019). All individual scores are statistically significant ($p < 0.05$), and no multiple comparison correction is applied.

# 3 Results and Discussion

**CORnet is the best among supervised models** CORnet-S, the current best model of the visual brain according to the Brain-Score benchmark (Schrimpf et al., 2018), outperforms both AlexNet and ResNet-50 on the 92-images dataset, except for MEG early interval, which is best explained by ResNet-50 (block2) (Table 3). On the 118-images dataset, CORnet-S (V2) is also the best model for fMRI but is outperformed by AlexNet (conv2) on MEG data. These results are consistent with the Brain-Score benchmark and suggest that CORnet-S is a competitive model overall despite being outperformed on data from specific cortical areas such as V4 (Kubilius et al., 2019).

**Predictive coding yields representations similar to brain data** The predictive coding model used in our experiments does not rely on labeled data and learns just by minimizing the error between predicted and actual sensory data. Most importantly, it is trained on a dataset of videos portraying activities such as *cooking*, *walking*, and *dancing*, which differ significantly from the carefully curated 92 and 118-images datasets used to measure brain activity. Despite those differences, the model trained on just 1 hour of videos from the KITTI dataset and 7 million parameters outperforms an AlexNet model ($\approx 61$ million parameters, see comparison in Table 2) in terms of correlation to both fMRI and MEG data (92-images dataset, Table 3) and fMRI data (118-images dataset, Table 4).

Table 2: The models used in the experiments. Layer count is given by the "number of convolutions and fully-connected layers along the longest path of information flow" (Kubilius et al., 2018).

|  | AlexNet | CORnet-S | ResNet-50 | PredNet-5 | PredNet-4 |
|---|---|---|---|---|---|
| # layers | 8 | 15 | 50 | 15 | 12 |
| # parameters | $\approx 61M$ | $\approx 53M$ | $\approx 25M$ | $\approx 13M$ | $\approx 7M$ |

**Scaling unsupervised learning improves correlation with brain data** The noise normalized correlation scores (Tables 3 and 4) shows that learning more about how events unfold over time improves correlation scores. Correlation to brain representation continues to improve as we train the PredNet-4 and PredNet-5 models with up to 6 hours of videos from the Moments in Time dataset, except for IT scores, which do not exhibit a clear trend. Since PredNet-5 input has higher dimensionality ($256 \times 256$) and more layers, more experiments would be needed to separate the impact of the amount of training data and model size on the correlation scores.

**Why PredNet may be better than competing supervised baselines?** The PredNet architecture fulfills several of the desired properties a brain-like model should have, such as few layers, single canonical circuitry in all areas, and recurrency (Kubilius et al., 2018). Additionally, PredNet is designed to process spatiotemporal sensory data, such as sequences of video frames or audio spectrograms, which we argue is another crucial requirement for models that approximate the brain architecture. The use of recurrence in CORnet and other convolutional models (Nayebi et al., 2018) has been shown to capture neural dynamics, but still, they cannot "fabricate" real-world dynamics from static images. For instance, it is known that clustering of animate/inanimate objects is captured from neural activity in the IT area (Khaligh-Razavi & Kriegeskorte, 2014). We believe the emergence of those semantic divisions requires experiencing how the objects/entities behave in space and time.

Table 3: Correlation scores for model layers with best average correlation. Evaluation is performed on the **92-images dataset** in terms of models RDM noise normalized squared correlation (Spearman) to fMRI and MEG RDMs. All PredNet features are from recurrent timestep 10.

| | | Noise Normalized $R^2$ (%) | | | |
| | | fMRI | | MEG | |
| Model Name | Pre-training | EVC | IT | Early | Late |
|---|---|---|---|---|---|
| AlexNet (conv2) | ImageNet | 16.79 | 7.11 | 4.88 | 15.09 |
| AlexNet (conv4) | ImageNet | 13.98 | 10.66 | 2.72 | 16.68 |
| ResNet-50 (block2) | ImageNet | 10.80 | 1.28 | 12.46 | 8.33 |
| ResNet-50 (block4) | ImageNet | 12.45 | 10.02 | 5.29 | 12.60 |
| CORnet-S (IT) | ImageNet | 22.27 | 14.19 | 9.47 | 20.26 |
| PredNet-4 (layer 1) | random weights | 5.25 | 1.09 | 2.33 | 3.81 |
| PredNet-4 (layer 4) | KITTI (1h) | 39.90 | 13.13 | 29.52 | 20.16 |
| PredNet-4 (layer 4) | KITTI + Moments (4h) | 47.03 | **15.27** | 32.22 | 16.78 |
| PredNet-5 (layer 3) | Moments (6h) | **51.48** | 14.52 | **40.35** | **22.89** |

Table 4: Correlation scores for model layers with best average correlation. Evaluation is performed on the **118-images dataset** in terms of models RDM noise normalized squared correlation (Spearman) to fMRI and MEG RDMs. All PredNet features are from recurrent timestep 10.

| | | Noise Normalized $R^2$ (%) | | | |
| | | fMRI | | MEG | |
| Model Name | Pre-training | EVC | IT | Early | Late |
|---|---|---|---|---|---|
| AlexNet (conv2) | ImageNet | 11.20 | 1.91 | 20.18 | 8.45 |
| ResNet-50 (block2) | ImageNet | 9.93 | 4.38 | 8.64 | 2.53 |
| CORnet-S (V2) | ImageNet | 13.23 | 5.21 | 12.68 | 3.94 |
| PredNet-4 (layer 1) | random weights | 8.60 | **9.77** | 2.42 | 3.00 |
| PredNet-4 (layer 3) | KITTI (1h) | 17.14 | 6.93 | 20.15 | 7.74 |
| PredNet-4 (layer 4) | KITTI + Moments (4h) | 22.93 | 4.94 | 44.48 | **18.24** |
| PredNet-5 (layer 3) | Moments (6h) | **24.73** | 5.20 | **44.68** | 15.20 |

## 4   Final remarks and future directions

In this paper, we report empirical results suggesting that unsupervised predictive coding models may be a better starting point than supervised image recognition models to explain visual brain data. While our results are still far from the noise ceilings, there is still much room for improvement by scaling the model architecture and training data (AlexNet has about nine times more parameters than the PredNet-4 model). Many of the current state-of-the-art models in machine learning are based on large-scale unsupervised pre-training of models with hundreds of millions of parameters (Devlin, Chang, Lee, & Toutanova, 2018). Whether the predictive coding approach would benefit from such an extreme scaling approach is an open research question.

The correlation scores could also be improved by fine-tuning the model, combining features of different layers, performing feature selection, or even manually annotating the provided RDMs according to the semantic categories such as animate/inanimate and face/face-non visual stimuli. These approaches were successfully applied in previous work by Khaligh-Razavi and Kriegeskorte (2014) and by the leading entries of the Algonauts Challenge [1]. However, our goal is not to outperform all the models in a specific, small dataset but to evaluate the capacity of transferring representations learned from (potentially large) unlabeled out-of-domain datasets. In the future, we plan to test predictive coding representations in more comprehensive *brain-likeness* benchmarks such as the Brain-Score (Schrimpf et al., 2018).

---

[1]The submission reports are available at the Algonauts Challenge website: `http://algonauts.csail.mit.edu/challenge.html`

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
