# OpenReview forum: "Learning to predict visual brain activity by predicting future sensory states"
_NeurIPS.cc/2019/Workshop/Neuro_AI — Real Neurons & Hidden Units @ NeurIPS 2019 Poster_

### Official Review · AnonReviewer2 · 2019-09-23
**Training of predictive networks on unlabeled video data appears to increase similarity to brain data**

**Clarity:** 4

**Comment:**

It does not seem like predictive coding is the main thing going on in V1 (Stringer et al., Science 2019), so I’d be curious how the authors think that should be taken into account in the future.

Typo line 24 – “Moreover, we show that as (we) train the model…”
Typo line 87 – “Second, the model does not rely on labeled data and learn(s)”

**Category:**

Common question to both AI & Neuro

**Clarity Comment:**

Results were presented quite clearly, although datasets and methods rely entirely on previously published work, such that digging into previous work on PredNet and the Algonauts project was necessary for a full understanding.

**Evaluation:**

3: Good

**Importance:**

3: Important

**Importance Comment:**

I believe the concept of using predictive coding and unlabeled video data to train convnets is a great idea. However, the contribution of the authors does not appear to extend beyond combining existing data sets with existing network architectures.

**Intersection:**

4: High

**Intersection Comment:**

The question of how the visual world is represented in the brain is an essential question in neuroscience as well as for building successful machine learning techniques for artificial vision.

**Rigor Comment:**

The work is lacking a discussion of the most recent work in the similarity of visual processing in convnets to brain data, which incorporate recurrence into convnets (Nayebi et al. 2018, Kubilius et al. 2018 and 2019), thereby potentially allowing for similar behavior as a PredNet. How would you expect those networks to perform when trained on unlabeled video data?

It would have been useful to put these in context of the results of the algonauts contest, which pitched supervised methods such as Alexnet against user-submitted content. Does PredNet outperform other user-submitted models?

For this result to be convincing, I would like to see some reasons why the authors think PredNet is outperforming previous models. For example, is there something different about the feature maps that support this? What precisely about predictive coding makes the similarity to brain data expected?

**Technical Rigor:**

3: Convincing

---

> ### Author Response · Authors · 2019-10-25
> **AnonReviewer2 reply**
>
> Thank you for the corrections and for pointing out the invaluable related work. Please find my replies below.
>
> =====
>
> >>> The work is lacking a discussion of the most recent work in the similarity of visual processing in convnets to brain data, which incorporate recurrence into convnets (Nayebi et al. 2018, Kubilius et al. 2018 and 2019), thereby potentially allowing for similar behavior as a PredNet. How would you expect those networks to perform when trained on unlabeled video data?
>
> To address this omission, I included results for the CORnet-S model (Kubilius et al. 2018). Overall, CORnet-S outperforms the other supervised models, but still gives lower correlation scores than PredNet. It would be interesting to check if these results remain consistent on different benchmarks such as the Brain-Score. Comments are now included in the "Results and Discussion" section.
>
> Nayebi et al. 2018 and Kubilius et al. 2018, 2019 did a quite interesting investigation of the effects of recurrence to explain brain data. However, their overall architecture is still mostly linear/feedforward (see Nayebi et al. 2018, Figure 1) and would require a significant redesign to be trained on unlabeled videos. While I believe the ConvLSTM plays an important role in PredNet, the key ingredient is probably the top-down generative model that gives the representations used in our experiments (R_l representation units in Figure 1, Lotter et al. 2016). It would be interesting though to see if the "biologically-realistic" ConvRNN cells by Nayebi et al. 2018 would improve PredNet performance.
>
> =====
>
> >>> It would have been useful to put these in context of the results of the algonauts contest, which pitched supervised methods such as Alexnet against user-submitted content. Does PredNet outperform other user-submitted models?
>
> PredNet does not outperform the best submissions. The reason is that the leading entries apply clever methods to leverage the available 92 and 118-images training datasets, including fine-tuning, manual annotation using semantic categories, and feature selection (an example report: "https://arxiv.org/abs/1907.02591"). My attempts to fine-tune PredNet using the static images from the training set resulted in poor generalization on the test set, which I believe is caused by the lack of temporal information.
>
> In this work, I do not seek to beat benchmarks in specific fMRI/MEG datasets, but instead to compare different architectures and inductive biases in terms of similarity to brain activity, without in-domain fine-tuning.
>
> =====
>
> >>> For this result to be convincing, I would like to see some reasons why the authors think PredNet is outperforming previous models. For example, is there something different about the feature maps that support this? What precisely about predictive coding makes the similarity to brain data expected?
>
> The PredNet architecture fulfills several of the desired properties a brain-like model should have, such as few layers, single canonical circuitry in all areas, and recurrency (Kubilius et al. 2018). Additionally, PredNet is designed to process spatiotemporal sensory data, such as sequences of video frames or audio spectrograms, which I argue is another crucial requirement for models that approximate the brain architecture.  For instance, it is known that clustering of animate/inanimate objects is captured from neural activity in the IT area. I believe the emergence of those semantic divisions requires experiencing how the objects/entities behave in space and time.
>
> =====
>
> >>> It does not seem like predictive coding is the main thing going on in V1 (Stringer et al., Science 2019), so I'd be curious how the authors think that should be taken into account in the future.
>
> Stringer et al., 2019 report that behaviorally related activity is captured as early as V1 in mice. It is not clear to me how their results would be incompatible with the general framework of predictive coding. PredNets can learn representations for video and audio spectrograms (as I demonstrate in previous work: "https://github.com/thefonseca/msc-project/raw/master/dissertation.pdf"). As long as there is some suitable way to encode "efference" copies of motor commands, I see no reason why a PredNet could not account to some extent for the time-varying behavioral data. I would be glad if the reviewer can expand on that.

---

### Official Review · AnonReviewer1 · 2019-09-26
**Prediction-based network is similar to neural data; more detailed analysis would be helpful.**

**Clarity:** 4

**Comment:**

The authors present evidence that an unsupervised, predictive-coding model of vision is more correlated to neural data in its responses than popular, supervised models. This is further evidence that supervised feedforward models fail to capture something substantial about natural vision, although the particular predictive implementation described here might or might not be descriptive of reality. Without detailed insight into the similarities, it is difficult to evaluate whether the similarities are persuasive.

While the authors report similarity between PredNet representations and data as compared to feedforward architectures, there’s a question that seems to be unaddressed — which may be technically difficult to address fully, but is necessary to understand at least qualitatively: is the performance of the predictive network, at the volume of training data used in this study, comparable to the performance of the feedforward networks? An answer to this question in either direction would not detract from the results presented in the study, but would clarify what existing models have and have not captured about image processing in the brain.

**Category:**

AI->Neuro

**Clarity Comment:**

The paper poses an interesting question and develops it clearly. The empirical results reported are not straightforward to interpret.

**Evaluation:**

3: Good

**Importance:**

3: Important

**Importance Comment:**

The authors seek to develop unsupervised training models that exhibit image responses correlated to observed fMRI and MEG activity. This work fits into a tradition of similar inquiries and shows that a predictive, unsupervised network can exhibit higher correlation to neural activity than a supervised convolutional network.

**Intersection:**

4: High

**Intersection Comment:**

The paper asks whether artificial neural networks can be useful models of brain activity, and therefore sits comfortably at the interesection of neuroscience and artificial intelligence.


**Rigor Comment:**

The use of multiple datasets contributes to the rigor of the paper. While the similarity measures reported in the paper are based in literature, as reported they are quite opaque, and more detailed analysis would be needed to be persuasive. More detailed reporting of similarity (e.g. example trials or time-courses, alternate criteria) could be helpful.


**Technical Rigor:**

2: Marginally convincing

---

### Official Review · AnonReviewer3 · 2019-09-27
**Representations from unsupervised models are correlated with brain activity, details missing**

**Clarity:** 3

**Category:**

AI->Neuro

**Clarity Comment:**

This paper is well written but misses many details that are important to understand everything that was done (see other comments).

**Evaluation:**

3: Good

**Importance:**

3: Important

**Importance Comment:**

This paper shows data that refutes a previous result that representations from unsupervised models are not good at predicting visual areas (mainly IT). Instead, the authors find that a model trained to predict the next frame in a video is more correlated with visual areas.

Minor: Which unsupervised models did the 2014 paper use? Should we call PredNet self-supervised?

Only using ResNet and AlexNet are not enough as baselines as a lot of recent work has been done in this area.

**Intersection:**

5: Outstanding

**Intersection Comment:**

This paper is exactly at the intersection of (cognitive) neuroscience and AI.

**Rigor Comment:**

The paper appears sound. RSA with Spearman correlation is widely used in the field but I don't think that it is the best approach to relate different representations as its properties are not theoretically analyzed. There are other theoretically derived similarity measures or other approaches such as encoding models that I believe lead to more interpretable results. It is also not mentioned how the noise ceiling was estimated and accounted for.

The authors mention all results are statistically significant. Does that mean that the individual correlations are higher than chance or the bolded correlations are significantly higher than the others? Has multiple comparison correction been made? All details are useful and there is space for them.

**Technical Rigor:**

3: Convincing

---

> ### Comment · ~Marcio_Fonseca1 · 2019-11-05
> **AnonReviewer3 reply**
>
> Thanks for your review. Please find below the answers to your comments.
>
> =====
>
> >>> Minor: Which unsupervised models did the 2014 paper use?
>
> The unsupervised models are:
> - VisNet (Wallis & Rolls, 1997)
> - HMAX (Riesenhuber & Poggio, 1999)
>
> I did not consider these models since they were shown by Khaligh-Razavi & Kriegeskorte (2014) to be outperformed by AlexNet.
>
> =====
>
> >>> Should we call PredNet self-supervised?
>
> Although I follow Lotter et al. (2016) and call PredNet "unsupervised," I would agree that self-supervised would be more precise as the learning objective consists in using part of data (up to the current frame) to predict another part (next frame).
>
> =====
>
> >>> Only using ResNet and AlexNet are not enough as baselines as a lot of recent work has been done in this area.
>
> That's true. To address this omission, I included results for the CORnet-S model (Kubilius et al. 2018) and a short discussion addressing the model by Nayebi et al. (2018). Please refer to my reply to AnonReviewer2 below.
>
> =====
>
> >>> There are other theoretically derived similarity measures or other approaches such as encoding models that I believe lead to more interpretable results
>
> I agree that a more comprehensive set of benchmarks would make the results more convincing and interpretable. I propose to evaluate the features on the Brain-Score (Schrimpf et al., 2018) in the "Final remarks and future directions" section.
>
> =====
>
> >>> It is also not mentioned how the noise ceiling was estimated and accounted for.
>
> The noise ceiling is the average of the correlations of each subject’s RDMs to the average RDM across 15 subjects. Thus, the subject-averaged RDM is assumed to be the best estimate of an RDM generated by an ideal model (Cichy et al., 2019). This description is now added to the "Methods > Evaluation" section.
>
> =====
>
> >>> The authors mention all results are statistically significant. Does that mean that the individual correlations are higher than chance or the bolded correlations are significantly higher than the others? Has multiple comparison correction been made?
>
> Statistical significance refers to individual correlation scores. No multiple comparison correction is applied. This clarification is now added to the "Methods > Evaluation" section.

---

### Decision · Program_Chairs · 2019-10-02

Accept (Poster)